# Genome-Wide Variation Analysis of Four Vegetable Soybean Cultivars Based on Re-Sequencing

**DOI:** 10.3390/plants11010028

**Published:** 2021-12-23

**Authors:** Xiaomin Yu, Xujun Fu, Qinghua Yang, Hangxia Jin, Longming Zhu, Fengjie Yuan

**Affiliations:** Institute of Crop and Nuclear Technology Utilization, Zhejiang Academy of Agricultural Sciences, Hangzhou 310021, China; fuxj@zaas.ac.cn (X.F.); yangqh@zaas.ac.cn (Q.Y.); jinhx@zaas.ac.cn (H.J.); zlmsllzly@163.com (L.Z.)

**Keywords:** soybean, genetic diversity, molecular marker, gene function, structure variation, vegetable soybean

## Abstract

Vegetable soybean is a type of value-added specialty soybean, served as a fresh vegetable or snack in China. Due to the difference from other types, it is important to understand the genetic structure and diversity of vegetable soybean for further utilization in breeding programs. The four vegetable cultivars, Taiwan-75, Zhexiandou No. 8, Zhexian No. 9 and Zhexian No. 10 are popular soybean varieties planted in Zhejiang province, and have large pods and intermediate maturity. The clustering showed a close relationship of these four cultivars in simple sequence repeat analysis. To reveal the genome variation of vegetable soybean, these four improved lines were analyzed by whole-genome re-sequencing. The average sequencing depth was 7X and the coverage ratio of each cultivar was at least more than 94%. Compared with the reference genome, a large number of single-nucleotide polymorphisms, insertion/deletions and structure variations were identified with different chromosome distributions. The average heterozygosity rate of the single-nucleotide polymorphisms was 11.99% of these four cultivars. According to the enrichment analysis, there were 23,371 genes identified with putative modifications, and a total of 282 genes were related to carbohydrate metabolic processes. These results provide useful information for genetic research and future breeding, which can facilitate the selection procedures in vegetable soybean breeding.

## 1. Introduction

Soybean (*Glycine max*) is a legume species native to China which has spread all over the world, with a history of more than 5000 years [1]. Due its rich protein, oil, isoflavone, unsaturated fatty acid, vitamins and other nutrients, soybean has become an important crop as both food and forage [2,3,4]. Due to the different harvesting times, it can be divided into vegetable type (as fresh vegetable for consumption) and grain type (as raw materials for processing) [5,6]. Vegetable soybean, also named as “Maodou” in China, “Edamame” in Japan, and “Green vegetable soybean” in North America, is a kind of food-grade specialty soybean that could be served as a fresh vegetable, snack, or be used for cooking [7,8].

Vegetable soybean is typically harvested during the R6 and R7 growth stages, when the color of pods is still green and seeds comprise about 85% of the total weight [3,9]. The key features are big green pods (longer than 5.0 cm, wider than 1.3 cm), large seeds with green seed coats (fresh weight of 100 seeds over 70 g), high sucrose content, low indigestible oligosaccharides, smooth texture and good flavor [7,10,11,12]. Vegetable soybean can either be sold fresh, as pods on the stems, stripped pods, shelled beans, or sold as a frozen or canned product [9,13]. Vegetable soybean has become a very popular vegetable crop worldwide because of its multiple health benefits as well as its economic importance [6,8].

With an increasing emphasis on the collection, evaluation and utilization of germplasm resources, some studies have been carried out to reveal the genetic structure and diversity of vegetable soybean cultivars and landraces [7,14]. Mimura et al. [15] applied 17 simple sequence repeat (SSR) markers to study the genetic diversity among 130 edamame accessions from China, Japan and the United States. The clustering showed that the patterns of SSR diversity in edamame can generally distinguish maturity classes and testa color. Dong et al. [7] analyzed the genetic structure and diversity of 100 vegetable soybean accessions planted in China using 53 SSR markers. These germplasms could be divided into 11 subgroups, and each group showed great consistency with the original seed coat colors or pedigrees. Genetic variation is the main determinant of phenotypic differences, which have been widely used in population structure and linkage disequilibrium analysis, high-throughput genetic typing and marker-assisted breeding.

Whole-genome re-sequencing is a second-generation sequencing technology, which can realize the parallel sequencing of different individual genomes. Through re-sequencing and comparison with the reference genome (Wm82.a2.v1), we can identify genomic variations, including single-nucleotide polymorphisms (SNPs), insertion/deletions (InDels), and structure variations (SVs) [16,17,18]. Lam et al. [19] re-sequenced the genomes of 31 soybean accessions (including 14 cultivated and 17 wild soybeans): a high level of linkage disequilibrium was identified in the soybean genome, which was distinct from other crops. The level of genetic diversity in wild soybeans was higher than that in cultivated soybeans. Maldonado dos Santos et al. [20] re-sequenced the genomes of 28 Brazilian soybean cultivars released over the last 50 years. A large number of allelic and structural variations were identified that can be used in marker-assisted selection for crop improvement. The Brazilian soybean germplasm remained narrow due to the large number of genome regions with low diversification.

In this study, we re-sequenced the four vegetable soybean cultivars that are planted across large areas of Zhejiang province in China. The sequencing data were used to evaluate genomic variations of these four cultivars. Although the clustering of SSR analysis suggested the similar pedigree of these four cultivars, they still showed a different distribution of genetic variations on each chromosome. Additionally, a group of genes was identified with putative modifications that were related to sugar and starch metabolism. These results will help us understand the divergence between vegetable and grain soybeans and further improve our breeding strategies.

## 2. Results

### 2.1. Character Comparison

Agronomic traits play a critical role in the acceptability and marketability of vegetable soybean; the main agronomic characters of the four cultivars (Taiwan-75, Zhexiandou No. 8, Zhexian No. 9 and Zhexian No. 10) were investigated in both 2019 and 2020 (Table 1). The growth durations of these four cultivars were between 84 and 85 d, with no significant differences. Therefore, these four cultivars could be classified into an intermediate maturity group of the southern spring soybeans in Zhejiang. The plant heights of these four cultivars were between 35 and 45 cm, which is significantly lower than those of grain soybeans recorded (commonly between 45 and 55 cm) grown in Zhejiang. The flower and fresh pod colors of these four cultivars were white and green, respectively. Although all cultivars had the big pod size and great seed weight, the cultivar Zhexian No. 9 had the longest pod length (6.4 cm) with the greatest fresh 100-seed weight (88 g). Furthermore, the contents of soluble sugar and starch both suggested good-quality characteristics of fresh seeds (Table 1). Based on these characters, all the four cultivars could meet the superior standard of vegetable soybean in China (Figure 1).

### 2.2. SSR Analysis

It is informative and effective to use simple sequence repeat (SSR) markers as a tool to analyze genetic diversity. Here, the 40 SSR markers provided unambiguous fragments and gave a total of 234 alleles across the four cultivars (Taiwan-75, Zhexiandou No. 8, Zhexian No. 9 and Zhexian No. 10) (Appendix A). The least polymorphic marker, satt588, and the most polymorphic marker, satt005, amplified 2 and 11 alleles, respectively, with an average of 5.85 alleles per locus. The unweighted pair group method using arithmetic mean (UPGMA) cluster analysis showed that the genetic similarity coefficient of these four cultivars was higher than 0.83 (Figure 2). The relationship of Taiwan-75 and Zhexiandou No. 8 was close compared with the other cultivars. These four cultivars showed a close relationship, and were confirmed to have a similar pedigree.

### 2.3. Variation Detection

Using Illumina high-throughput sequencing technology, the whole genomes of the four cultivars (Taiwan-75, Zhexiandou No. 8, Zhexian No. 9 and Zhexian No. 10) were re-sequenced with an average coverage depth of 7X. The average percentage of reads properly mapped to the genome was 93.16%, with a coverage ratio of at least 94%, and the average proportion of high-quality reads with quality scores greater than 30 was 92.24% (Appendix A). The genetic variations, including SNPs, InDels and SVs, were identified with a different distribution on the chromosome through comparison with the reference genome (Figure 3 and Appendix A).

An average of 1,413,746 SNPs were identified in these four cultivars. Most of the SNPs were homozygous, with an average of 1,243,199 (Table 2). Nonetheless, 11.99% of the SNPs were heterozygous and the cultivar Taiwan-75 possessed the greatest number of heterozygous SNPs. On average, 640,456 SNPs were detected in intergenic regions. We also found an average of 56,076 SNPs in exons, 146,721 SNPs in introns and 26,482 SNPs in untranslated coding regions (Appendix A). The SNP variation could be classified as transition and transversion and the average ratio of transition to transversion was 1.86, which was consistent with that observed in other soybean studies [17].

On average, 317,644 InDels were detected in the genome among these four cultivars, 1.33% of which were distributed in the coding region and in accordance with the proportion reported in other soybean studies [20]. Approximately 94,503 of the InDels were in the intergenic regions, 43,654 were in introns, 4223 in exons and 10,241 were in untranslated coding regions (Appendix A). The heterozygous to homozygous ratio for each cultivar was above 85% for InDels, which was different from that observed for SNPs.

### 2.4. Function Annotation

According to these four cultivars (Taiwan-75, Zhexiandou No. 8, Zhexian No. 9 and Zhexian No. 10), there were 59,495, 50,983, 56,451 and 57,373 SNPs identified in the coding regions, respectively. Additionally, 32,662, 27,865, 30,757 and 31,250 of them were non-synonymous mutations, which were identified in important regions of 15,338, 13,939, 14,709 and 14,794 genes, respectively (Table 3). The similar analysis of non-synonymous mutations in the coding regions was also conducted to identify 4557, 3670, 4279 and 4387 small InDels, respectively. These InDels of the four cultivars were non-synonymous modifications detected in 4073, 3384, 3884 and 3933 genes, most of which were in exons (Table 3).

Combined with those results, a total of 23,371 genes were identified with putative modifications in these four cultivars compared with the reference genome (Figure 4). The contents of sugar and starch were important quality characteristics of vegetable soybean. Based upon the annotation, 282 genes were associated with the carbohydrate metabolic process and 16 genes were related to the sucrose metabolic process (Appendix A). Among them, three genes were related to carbohydrate transport and metabolism, including *Glyma.01g211000*, *Glyma.05g056300* and *Glyma.17g138500*. Moreover, 10 genes were annotated as related to the starch metabolic process. Additionally, one of the genes (*Glyma.01g124200*) was associated with the GYF-domain protein, possibly playing a role in protein–protein interactions [21].

## 3. Discussion

The cumulative effect of breeding selection resulted in significant differences in both morphological and quality characteristics between vegetable type and grain type soybeans [5,6,7]. To reveal the genome differences between these two types, two vegetable cultivars and two grain cultivars were analyzed by whole-genome re-sequencing simultaneously [22]. The SNP rates of grain and vegetable soybeans were 12.24% and 10.97%, respectively. Thus, the rate of vegetable soybean was lower than that of the grain type. Here, the average heterozygosity rate was 11.99% in these four cultivars. These results were consistent with the previous similar studies [7,8,15], and showed the difference between vegetable and grain soybeans at the genome level. It is speculated that the SNP heterozygosity is closely related to the parental characteristics. Vegetable soybean is only one special part of the soybean germplasms, and its genetic variation is somewhat different from the grain type [20,23]. The range of parents available for the selection of grain soybean is much larger than that of vegetable soybean. Additionally, the breeding of vegetable soybean needs to introduce more abundant germplasm resources.

Although the four cultivars were presumed to have a similar pedigree, their pod length and seed weight were still statistically different. More than 10 SNPs associated with seed size and weight were reported across different chromosomes for grain soybean [24]. Multiple SNPs associated with seed weight were also identified across two different chromosomes for vegetable soybean [25,26]. Seed size and weight are important characteristics for vegetable soybean production. More SNPs associated with these seed traits could be observed; the vegetable soybean germplasms were much more diverse in China than other areas [7,15].

Taste and flavor are most important in vegetable soybean breeding; therefore, the breeding lines with high sucrose contents and good flavors are more carefully preserved in the selection process [7,15]. In this study, 16 and 10 genes were associated with sucrose and starch metabolic processes, respectively, some of which might play a role in sucrose and starch metabolism. A number of SNPs are reportedly associated with sucrose and protein contents [8,26,27]. Once confirmed, the differences could provide an insight into breeding selection of vegetable soybean as well as the loss of function of genes that may have a key role in grain soybean. However, more detailed studies are needed to verify the functions of these modified genes, especially those related to sucrose and starch metabolism.

The frequent usage of limited elite lines may lead to a relatively narrowed genetic background of vegetable soybean [20,21]. The development of related markers and the mining of specific genes can provide marker resources for molecular-marker-assisted breeding and further reveal the genome composition of the two soybean types [27,28]. This will have an important guiding significance for the classification of vegetable soybean breeding, and can effectively improve the directivity of vegetable soybean breeding and promote the development of the vegetable soybean industry.

## 4. Materials and Methods

### 4.1. Plant Materials

The cultivar Taiwan-75 (Zhepin No. 271) was introduced to Zhejiang from Taiwan in 2000. The other three cultivars, Zhexiandou No. 8 (Zhenshen No. 2012001), Zhexian No. 9 (Zheshen No. 2015001) and Zhexian No. 10 (Minshen No. 2015004), were developed by the Zhejiang Academy of Agricultural Sciences. The four improved cultivars were grown in a randomized complete block design in three replicates in early April 2019 and 2020 in the experimental field of Zhejiang Academy of Agricultural Sciences. For each replicate, three-row plots 5 m long and 0.4 m row-spaced were planted, with 10 plants grown per meter.

### 4.2. Character Investigation

According to the local planting habits, the pod samples were collected at the beginning of the maturity stage (R7) at each year. All plants within 2 m of the inner row per plot were cut from the bottom of the stem. These sampled plants were measured to determine the plant height, and pods were collected immediately. A random sub-sample of 100 two-seed pods was taken to determine the size and weight. Seeds were collected and weighed immediately for determining the fresh seed yield. The contents of soluble sugar and starch were measured according to the Chinese national standard methods (GB5009.8-2016 and GB5009.9-2016). The analyses were performed in triplicate and expressed on a fresh weight basis. Significant statistical differences were determined by one-way analysis of variance (ANOVA). Multiple comparisons were conducted using Tukey method with SPSS statistics 19.0 (IBM, Armonk, NY, USA).

### 4.3. SSR Analysis

Genomic DNA was isolated from these pods of each cultivar using CTAB method [29]. The simple sequence repeat (SSR) analysis was performed according to the previous reports [7,30]. Based upon the previous reports, 40 SSR markers which were evenly distributed on 20 soybean genetic linkage groups were used in the genotyping. A zero–one data matrix was created according to the absence or presence of alleles for each cultivar. The unweighted pair group method using arithmetic mean (UPGMA) cluster analysis was performed to construct a distance tree.

### 4.4. Sequencing and Variation Detection

The sequencing procedure was performed according to the standard protocol provided by Illumina [31,32]. After being qualified, DNA was fragmented by a mechanical method. Then, fragment purification, terminal repair, 3′ end plus A, and sequencing were used separately. Subsequently, agarose gel electrophoresis was used to separate the fragments and PCR was used to amplify the library. After quality inspection, the DNA was sequenced by Xten [33,34,35]. The quality of raw reads was evaluated, and the clean reads were obtained by filtering; these were compared with the reference genome sequence (Wm82.a2.v1) at SoyBase [16]. SNPs, InDels and SVs were detected using the Genome Analysis Toolkit (GATK) and functionally annotated using the SnpEff program [36,37,38].

## 5. Conclusions

Vegetable soybean is a kind of food-grade specialty soybean which has become a very popular vegetable crop. The findings of this study revealed both phenotypic and genotypic variation among four vegetable cultivars. The SNPs associated with seed size and sugar content were more relevant to vegetable soybean. Due to the genetic variation among cultivars, three genes were found to be related to carbohydrate metabolism. The identification of specific genes and the development of related markers could reveal the genome composition of vegetable soybean and thus improve vegetable soybean breeding.

## Figures and Tables

**Figure 1 plants-11-00028-f001:**
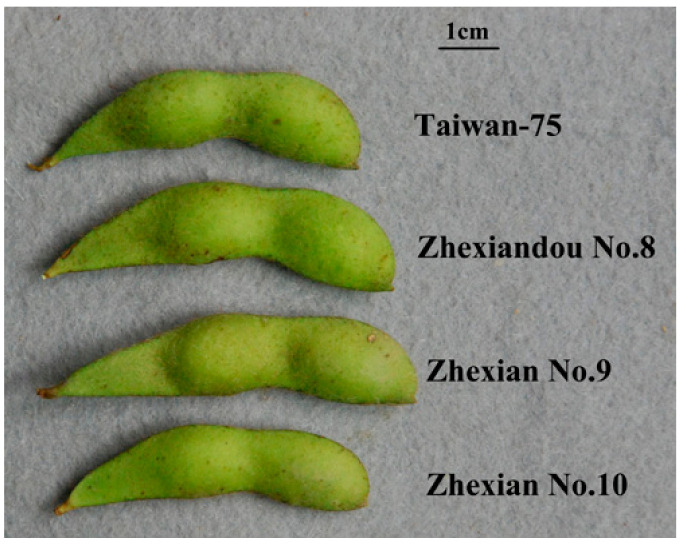
Fresh pods of the four cultivars.

**Figure 2 plants-11-00028-f002:**
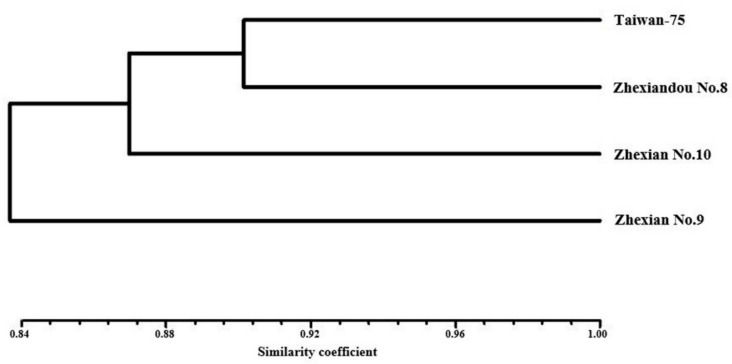
Clustering of the four cultivars based on the UPGMA analysis.

**Figure 3 plants-11-00028-f003:**
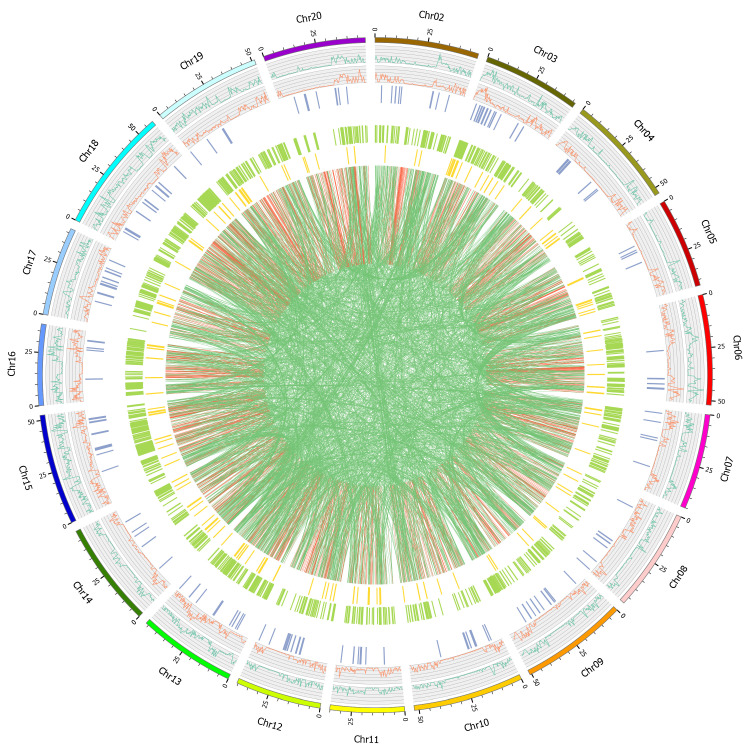
The distribution of the variations of Taiwan-75 on the chromosomes. The order from outside to inside: Chromosome, SNP distribution, InDel distribution, CNV, SV (INS, DEL, INV, ITX (red line), CTX (green line)) distribution (Unit M).

**Figure 4 plants-11-00028-f004:**
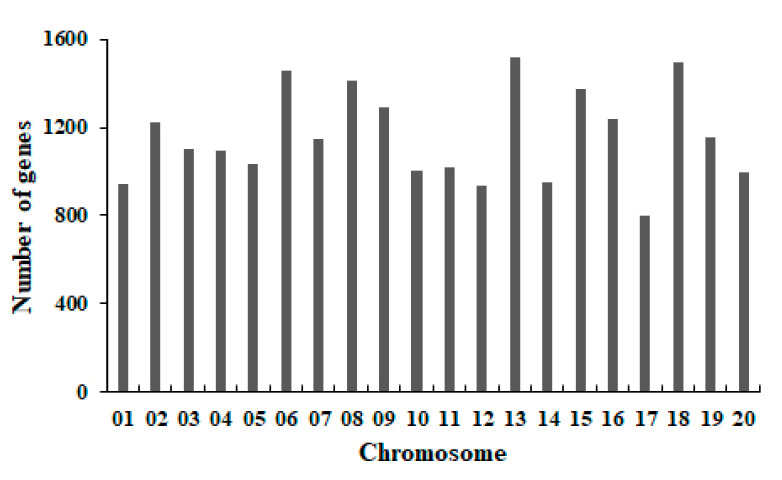
The gene distribution on each chromosome.

**Table 1 plants-11-00028-t001:** Means of some characters for the four cultivars over 2 years.

Cultivar Name	Growth Duration (d)	Plant Height (cm)	Pod Length (cm)	Pod Width (cm)	Fresh 100-Seed Weight (g)	Soluble Sugar Content (g/100 g)	Starch Content (g/100 g)
Taiwan-75	84 a ^1^	40 b	6.0 bc	1.4 a	83 bc	3.1 a	5.2 a
Zhexiandou No. 8	85 a	36 c	6.1 b	1.4 a	85 b	2.8 bc	5.1 ab
Zhexian No. 9	85 a	35 c	6.4 a	1.4 a	88 a	2.9 ab	4.8 c
Zhexian No. 10	85 a	44 a	5.9 c	1.4 a	80 c	2.6 c	4.9 bc

^1^ Means in a column followed without a common letter are significantly different (*p* < 0.05).

**Table 2 plants-11-00028-t002:** The numbers of SNPs and InDels for each cultivar.

Cultivar Name	SNPs			InDels		
	Heterozygous	Homozygous	Total	Heterozygous	Homozygous	Total
Taiwan-75	206,410	1,339,245	1,545,655	307,256	39,494	346,750
Zhexiandou No. 8	132,889	1,143,801	1,276,690	252,400	24,069	276,469
Zhexian No. 9	166,980	1,280,183	1,447,163	291,262	32,883	324,145
Zhexian No. 10	175,910	1,209,567	1,385,477	287,939	35,274	323,213

**Table 3 plants-11-00028-t003:** The gene numbers of the variations for each cultivar.

Cultivar Name	Genes with Non-Synonymous SNPs	Genes with InDels	Genes with SV	Total
Taiwan-75	15,338	4073	275	16,828
Zhexiandou No. 8	13,939	3384	146	15,237
Zhexian No. 9	14,709	3884	215	16,186
Zhexian No. 10	14,794	3933	230	16,287

## Data Availability

The data generated in this study are included in this published article and its Appendix A.

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
