# Peer review of "Genome-Wide Variation Analysis of Four Vegetable Soybean Cultivars Based on Re-Sequencing"

_plants, 2021, doi:10.3390/plants11010028_

Round 1
Reviewer 1 Report
Authors have carried out studies on studying the genome wide variation in four vegetable soybean cultivars based upon next generation sequencing. The study involved the phenotypic characterization of these four cultivars and in silico identification of different pathway related genes.
Soybean is an interesting legume species having rich protein oil and different nutrients having medicinal value. It is also used in snacks industry now a day.
It is an interesting study but needs more efforts to make this paper suitable for publication.
Introduction
English language needs to be improved
More recent articles need to be cited supporting the present study
Needs to include the reference genome article and its findings
Material and methods
Methods to measure the soluble sugar and starch should be defined clearly.
Give reference for CTAB method
List of SSR markers should be provided with details
Results
Table 1: Couldn’t understand the a,b,c written along with the values
Figure 1: Are you trying to measure the size of the pod. Clarify this.
Line 1: replace bands with fragments or amplicons
Line 111-112: Reframe the sentence
Is the sequencing depth of 7X was sufficient enough to analyse the data?
Why only genes related to carbohydrate metabolic processes are mentioned if the objective of the study was to find out genome-wide variation?
Were SNPs classified into synonyms and non-synonyms?
Line 169-170: transcription of what?
Discussion
Line 184: Replace objects with materials
Line 243-274: whole discussion is written like results only. Though actual aim of this study to characterize genome wide variations obtained in various agronomically important traits. So just discussing how many different SNPs obtained does not throw any light on any innovation. This could be relevant if more vigorously agronomic important traits have been studied and discussed between vegetable and grain soybean.
Overall comments
It is not well written, language needs lot of improvement. Some of the sentences need re-casting.
Authors must conclude in a short closing paragraph based on their results differentiating vegetable and grain soybean. Nothing about genetic diversity has been discussed taking their origin of evolution. More recent articles needs to be added in discussion part to support your study.
Discussion should be written critically with a focus on results.
Reviewer 2 Report
In Figure 1. Zhexiandou No. 8 should be shortened to Zhexian No. 8 to make it the same as No.0 and 10.
Figure 3 needs to be improved and separated into four figures to show the characteristics with one of them and put the rest of 3 as supplemental figures.
In line 131: An average of 1 413 746 SNPs were identified in these four cultivars, put a comma in the numbers, like 1,413,714, and format all the numbers in this format in the manuscript.
In discussion, the first paragraph was repeated from the introduction and can be deleted (line # from 175 to 184)
In the Conclusion section, it needs to rewrite as precisely as possible. Remember, the conclusion is NOT results, and these are the take-home message from your study.
